# The prion protein is not required for peripheral nerve de- and remyelination after crush injury

**Anna Henzi**, **Adriano Aguzzi***

Institute of Neuropathology, University of Zurich, Zurich, Switzerland

* adriano.aguzzi@usz.ch

## Abstract

The cellular prion protein (PrP) is essential to the long-term maintenance of myelin sheaths in peripheral nerves. PrP activates the adhesion G-protein coupled receptor Adgrg6 on Schwann cells and initiates a pro-myelination cascade of molecular signals. Because Adgrg6 is crucial for peripheral myelin development and regeneration after nerve injury, we investigated the role of PrP in peripheral nerve repair. We performed experimental sciatic nerve crush injuries in co-isogenic wild-type and PrP-deficient mice, and examined peripheral nerve repair processes. Generation of repair Schwann cells, macrophage recruitment and remyelination were similar in PrP-deficient and wild-type mice. We conclude that PrP is dispensable for sciatic nerve de- and remyelination after crush injury. Adgrg6 may sustain its function in peripheral nerve repair independently of its activation by PrP.

## Introduction

Schwann cells, the glial cells of the peripheral nervous system (PNS) interact with axons and the extracellular matrix during development, maintenance and repair of peripheral nerves [1, 2]. In response to injury, Schwann cells exhibit a remarkable plasticity, which is central to the capacity for regeneration in the PNS [3]. The study of repair processes after experimental nerve cut or crush injuries in transgenic animals has unraveled many of the molecular mechanisms and signaling pathways driving Schwann cell plasticity [4, 5]. After injury, Schwann cells undergo reprogramming to become repair Schwann cells. While repair Schwann cells reactivate some developmental signaling pathways, they also acquire a set of unique features which are not found in developing Schwann cells [6–9]. Signaling of the adhesion G-protein coupled receptor Adgrg6 (formerly called Gpr126) was first described to initiate developmental myelination in zebrafish [10] and mice [11]. Later, a contribution of Adgrg6 to macrophage recruitment and remyelination after nerve crush injury was demonstrated [12], indicating that Adgrg6 is also involved in PNS repair. Furthermore, Adgrg6 is required for responses of terminal Schwann cells and regeneration of neuromuscular junctions [13], which is crucial for the restoration of function after peripheral nerve injury. Finally, ageing conditional Adgrg6 knockout mice showed signs of hindlimb denervation [13], suggesting a role of Adgrg6 signaling in maintenance of peripheral myelin. This notion is supported by the development of a progressive demyelinating neuropathy of the PNS in mice devoid of the cellular prion protein

**Data Availability Statement:** All relevant data are within the paper and its Supporting information files.

**Funding:** AA is the recipient of a Swiss National Research Foundation Sinergia grant (CRSII5

183563). http://www.snf.ch/en/funding/
programmes/sinergia/Pages/default.aspx. The
funders did not play any role in study design, data
collection, analysis, preparation of the manuscript
or decision to publish.

**Competing interests:** The authors have declared
that no competing interests exist.

(PrP), an agonist of Adgrg6 [14, 15]. PrP is a glycophosphoinositol-anchored glycoprotein highly expressed in the nervous system [16] and mainly known for its role in prion diseases [17, 18]. The development of a chronic peripheral demyelinating neuropathy in PrP knockout mice has led to the identification of cellular PrP as necessary for myelin maintenance [15, 19]. Characterization of cell type-specific *Prnp* knockout mice revealed that PrP expressed by neurons but not by Schwann cells is required for peripheral myelin maintenance [15]. Neuronal PrP acts through Adgrg6 expressed on Schwann cells to activate pro-myelination signaling pathways [14]. While many phenotypes described in early generations of *Prnp* knockout mice have later been assigned to genetic confounders, the role of PrP in myelin maintenance has been confirmed in strictly co-isogenic C57BL/6J-*Prnp* knockout mice ($Prnp^{ZH3/ZH3}$) [20, 21].

The findings that PrP is an agonist of Adgrg6 and that Adgrg6 is involved in PNS repair raise the question whether PrP may also play a role in PNS repair. Here, we hypothesized that PrP may represent one of the ligands required to activate Adgrg6 after nerve injury. If so, PrP-deficient mice might develop similar defects in nerve regeneration as were reported in Adgrg6 knockout mice. While neuronal PrP expression is sufficient to support myelin maintenance, PrP from other cellular sources could contribute to remyelination and repair after injury. Therefore, we used $Prnp^{ZH3/ZH3}$ mice with global constitutive knockout of PrP for the experiments. To address our hypothesis, we performed nerve crush injuries in female $Prnp^{ZH3/ZH3}$ mice and compared the morphological and molecular stages of repair to wild type (WT) mice. Specifically, we assessed generation of repair Schwann cells, demyelination, recruitment of inflammatory cells and remyelination.

Contrary to our expectations, we detected no differences between $Prnp^{ZH3/ZH3}$ and WT mice in any of these processes. These findings suggest that PrP might be dispensable for peripheral nerve repair.

## Material and methods

### Mice

Breeding and maintenance of mice was performed in laboratory animal facilities (optimal hygienic conditions) at the University Hospital Zurich. Mice were housed in groups of 3 with unlimited access to food and water. Animal care and experiments were performed in accordance with the Swiss Animal Protection Law. All experiments were approved by the Veterinary Office of the Canton of Zurich (permit ZH168/2019). Our standard operating procedures allowed to repair surgical sutures one time in case of reopening. In our experience, female mice are less prone to wound reopening than male mice, and they can be regrouped after a period of single housing. Thus, we performed the experiment with female mice only.

### Nerve crush surgery

Surgery was performed on the right hind limb of two-month-old female WT and $Prnp^{ZH3/ZH3}$ mice as described previously [22, 23]. Briefly, mice were injected with buprenorphinum (Temgesic, 0.1 mg/kg of bodyweight) prior to surgery. The surgery was performed under isoflurane anaesthesia. Fur was removed with an electric trimmer and the sciatic nerve was exposed at the height of the hip by making a small incision. The sciatic nerve was exposed by blunt dissection and crushed by squeezing firmly for 30 seconds with a Dumont S&T JF-5 forceps (FST tools) at the height of the sciatic notch. The tissue was repositioned, and the wound was sealed with a suture. Mice were administered buprenorphinum for analgesia during the first 2 postoperative days. Sciatic nerves from crushed and contralateral side were harvested at 5, 10, 12, 16 and 30 days post crush.

## Nerve harvesting

Mice were sacrificed by cervical dislocation in deep anaesthesia. Sciatic nerves were embedded in the required fixation solution for morphological analysis or frozen in liquid nitrogen for protein analysis. After crush injury, the sciatic nerve distal to the crush site was divided in 3 parts (see S1 Fig). A segment distal to the crush site (3 mm) was used for electron microscopy (EM). The distal side of this segment was embedded facing the front of the block face for sectioning. Sections from 2 mm distal to the crush site were analysed by EM. The remaining sciatic nerve was cut in half and the more proximal segment was processed for immunofluorescence (IF) as described below. the nerve segment was positioned with the proximal side facing the front of the block face for sectioning. Thereby, the EM and IF images derived from a similar distance from the crush side. The most distal segment of the sciatic nerve was frozen in liquid nitrogen and used for protein analysis. On the contralateral side, the corresponding segments were collected in the same manner.

## Western blot analysis

Sciatic nerves were homogenized using stainless-steel beads in ice-cold lysis buffer (phosSTOP (Sigma, 4906845001) and protease inhibitor (Sigma, 11836153001) in RIPA buffer). Debris was removed by centrifugation of the lysates for 10 min at 10'000 g. Protein concentration was measured with BCA assay (Thermo Scientific). 10 µg protein for each sample was boiled in 4 x LDS (Invitrogen) at 95˚C for 5–10 min and loaded on a gradient 4–12% NuPAGE Bis-Tris gels (Invitrogen). After electrophoresis at 200 V, gels were transferred to PVDF membranes with the iBlot system (Life Technologies). Then, membranes were blocked with 5% SureBlock (LuBioScience GmbH, SB232010) in TBS-T. Incubation with primary antibodies was performed night at 4˚C. After three washes for 10 min, membranes were incubated with secondary antibodies coupled to horseradish peroxidase for 1 h at room temperature (RT). Membranes were then washed and developed with Crescendo chemiluminescence substrate system (Sigma, WBLUR0500). Signals were detected using a Fusion Solo S imaging system (Vilber). The FusionCapt Advance software was used for densitometry. For quantification, the protein levels of each sample were normalized to the respective loading control (actin or calnexin) and expressed as fold change to the average WT levels. Before incubating membranes with the next primary antibody, stripping was performed with Restore Western Blot Stripping Buffer (Thermofisher, 20163) for 15 min at RT and membranes were washed two times in TBS-T. Each lane in the blots and each point in the graphs represents one sciatic nerve from one mouse. Original, uncropped images are shown in (S2 Fig). The following primary antibodies were used for western blotting: rabbit antiGFAP (1:2000, Cell Signaling Technologies, 12389S), rabbit anti-c-Jun (1:1000, 9165s), rabbit anti-Calnexin (1:2000, Enzo, ADI-SPA-865-D), mouse anti-Actin (1:10'000, Milipore, MAB1501R). In addition, we used an in-house produced mouse monoclonal antibody (POM2) for detection of PrP [24]. The following horseradish peroxidase coupled secondary antibodies were used: anti-mouse IgG (1:10'000, Jackson Immuno Research, 115-035-003), anti-rabbit IgG (1:4000, Jackson Immuno Research, 111-035-003).

## Morphological analysis by EM

Sciatic nerves were immersed in 4% glutaraldehyde in 0.1 M sodium phosphate buffer pH 7.4 immediately after dissection and incubated at 4˚C overnight. Tissue was embedded in Epon using standard procedures. Further steps were performed as described [23]. Briefly, 99 nm sections were collected on ITO coverslips (Optics Balzers). Imaging for EM reconstruction of the entire sciatic nerve section was performed with either a Carl Zeiss Gemini Leo 1530 FEG or

Carl Zeiss Merlin FEG scanning electron microscope attached to Atlas modules (Carl Zeiss). Adobe Photoshop CS5 was used for image analysis. The g-ratio corresponds to the ratio between axon diameter and fibre diameter. The axon diameter was derived from the axon area. The myelin thickness was measured at two different locations of the myelin ring. The average myelin thickness was added twice to the axon diameter to obtain the fibre diameter. For g-ratio quantification, three different locations on the cross section were chosen and at least 100 fibres per sample were analysed. The number of intact appearing myelin profiles, remyelinated fibres and the area covered by myelin debris was assessed manually on the entire cross section. The investigator was blinded as to the genotype of the mice for all analyses.

## Immunofluorescence (IF)

Sciatic nerves were isolated as described above. The tissue was fixed in 4% paraformaldehyde overnight at 4˚C. Then, the tissue was incubated in 30% sucrose solution and frozen in OCT compound. Cross-sections of 8 μM thickness were cut at the cryostat. After drying at room temperature (RT), sections were incubated in blocking buffer (10% normal goat serum, 0.5% bovine serum albumin (BSA), 0.3% Triton X-100 in PBS) for 1 h at RT. Blocking buffer was diluted 1:1 with PBS for incubation with primary antibodies at 4˚C overnight. The following primary antibodies were used: rabbit anti-Ki67 (Abcam, ab15580, 1:1000), rabbit anti-c-Jun (Cell Signalling Technologies, 9165, 1:250), rabbit anti-S100 (Dako, Z0311, 1:250), mouse anti-S100 (Thermofisher, MA-12969, 1:50), rat anti-CD68 (Biorad, MCA1957, 1:200), rat anti-CD45 (BD Pharmagen 553078, 1:50), rabbit anti-Iba1 (Wako Bio-chemicals, 019–19741, 1:500). After three washes in PBS, slides were incubated with fluoro-phore-conjugated secondary antibodies (Alexa Fluor goat anti-rabbit 488 nm or 594 nm, goat anti-rat 594 nm or 647nm, goat anti-mouse 488 nm, Invitrogen) for 1 h at RT. Then, sections were washed in PBS, incubated with DAPI (1:10'000) to label nuclei for 10 min and mounted using Fluorescence mounting medium (Dako, S3023). Fluorescent images were recorded with a Leica SP5 Confocal Microscope. The original unaltered images from the fig-ures are provided in (S3 and S4 Figs).

For quantification of IF images, ImageJ (version 1.52) was used. To count DAPI, c-Jun and ki67-positive nuclei, nuclei were defined using the threshold function. Counting was per-formed on binary images. For each marker, the same ImageJ macro was applied to all images from one time point. The number of ki67-positive or c-Jun-positive nuclei was normalized to the total number of DAPI-positive nuclei in the field of view. Macrophages were quantified as the total CD68-positive area in relation to the S100-positive area. Iba1-positive cells and CD45-positive cells were counted manually using ImageJ. For each time point, three animals per group were analysed. For each mouse, the mean value from quantification of 2–4 images from separate nerve sections was used for statistical analysis. The investigator was blinded as to the time point and the genotype of the mice.

## Experimental design and statistical analysis

The sample size was chosen according to sample sizes used in previous publications performing western blot, IF and EM analyses of sciatic nerves [12, 25, 26]. Furthermore, it is in accordance with a sample size calculation based on the results from the CD68- and Iba1-immunohisto-chemistry studies reported in [12] (calculated with the G*power software version 3.1.9.7 with power 0.8, type I error rate 5%, effect size determined from [12] ranging from 2–6).

Statistical analysis was performed with GraphPad Prism software (version 8.4.2). We assumed normal distribution and equal variances of data but did not formally test this assump-tion due to small $n$ values. Unpaired two-tailed t-test was used for comparison of two groups.

For comparison of three or more groups, two-way ANOVA followed by Sidak's multiple comparison test was used and multiplicity adjusted p-values were reported. P-values below 0.05 were considered statistically significant. P-values are indicated in graphs as $^*$: $p < 0.05$. ns: not significant, $p > 0.05$. Error bars in graphs show SEM. No samples or data were omitted during the analyses. R (version 3.5.2) was used to generate the scatterplot visualizing the g-ratio analysis.

## Results

### PrP is not required for demyelination, repair Schwann cell generation and proliferation following nerve injury

To study the role of PrP in peripheral nerve regeneration, we performed sciatic nerve crush injuries in female $Prnp^{ZH3/ZH3}$ and WT mice and investigated the morphological and molecular stages of peripheral nerve repair. To avoid any interference of the demyelinating disease developed by ageing $Prnp^{ZH3/ZH3}$ mice, we performed the nerve crush injuries in mice at 2 months of age, when no morphological signs of demyelination can be detected [15, 21]. In particular, the number of endoneurial macrophages is not significantly increased in $Prnp^{ZH3/ZH3}$ mice at this age [21]. We crushed the sciatic nerve at the sciatic notch, and harvested the nerve segments distal to the injury site at 5, 10, 12, 16 and 30 days post crush (d.p.c.). Additionally, we collected the uninjured contralateral sciatic nerves as controls. To ensure comparability between the groups, the sections harvested for the various analyses were taken from the same distance from the crush site as depicted in (S1 Fig).

First, we assessed if Schwann cells in $Prnp^{ZH3/ZH3}$ mice were able to properly transdifferentiate to repair Schwann cells. We investigated c-Jun as a molecular marker of repair Schwann cells [6] by immunofluorescence (IF) and western blotting (Fig 1a–1d). At 5 d.p.c. a strong upregulation of c-Jun was detected both in WT and $Prnp^{ZH3/ZH3}$ mice. c-Jun levels decreased at later time points post injury. In uninjured contralateral nerves, c-Jun protein levels were low and nuclear c-Jun staining in IF was very faint. We did not detect any significant differences in c-Jun levels between WT and $Prnp^{ZH3/ZH3}$ mice at 5, 10, 16 and 30 d.p.c.

Next, we examined the expression of glial fibrillary acidic protein (GFAP). In the uninjured adult nerve, GFAP is mainly expressed by non-myelinating Schwann cells [27]. After injury, a strong upregulation of GFAP can be detected in sciatic nerves [28, 29], which is part of the Schwann cell injury response and contributes to efficient nerve regeneration [28, 30]. We quantified total sciatic nerve GFAP levels by western blotting (Fig 1c and 1d). GFAP levels were increased in crushed nerves compared to uninjured nerves, but no significant difference in GFAP levels was detected between $Prnp^{ZH3/ZH3}$ and WT mice at the investigated time points. At 5 d.p.c. GFAP levels in uninjured $Prnp^{ZH3/ZH3}$ mice appeared higher than in WT mice. By comparing GFAP expression in uninjured $Prnp^{ZH3/ZH3}$ and WT nerves (Fig 1e) we found that $Prnp^{ZH3/ZH3}$ mice had on average a two-fold higher baseline GFAP level when compared to WT mice (relative increase to WT GFAP level as quantified from Fig 1e: WT 1.00 ± 0.17 (6); $Prnp^{ZH3/ZH3}$ 2.05 ± 0.09 (6); mean ± SEM (n); $p = 0.0003$, unpaired t-test), which is in line with our previously published findings [31].

Ki67 staining in crushed sciatic nerves showed that proliferation was highest at 5 d.p.c. both in WT and $Prnp^{ZH3/ZH3}$ mice, and no significant difference in the proliferation index was detected between WT and $Prnp^{ZH3/ZH3}$ mice at any of the time points investigated (Fig 2a and 2b). To investigate which cell types contribute to proliferating cells, we performed co-staining with S100 for Schwann cells and CD68 for macrophages in nerve sections from 5 d.p.c. Most ki67-positive proliferating cells belonged to Schwann cells, whereas only few macrophages were positive for ki67. This indicated that macrophages did not significantly contribute to the population of proliferating cells at 5 d.p.c.

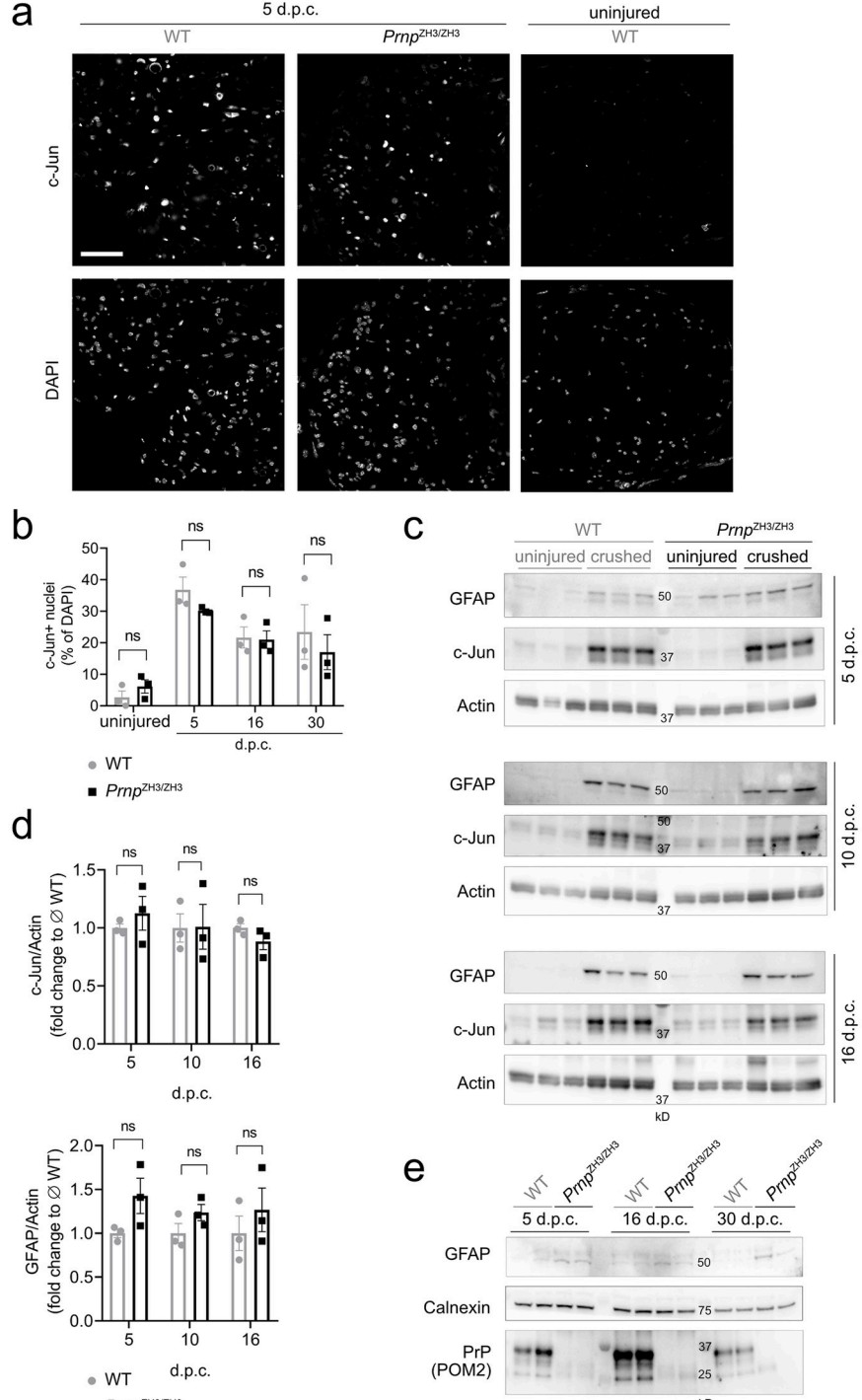

**Fig 1. Generation of repair Schwann cells.** (a) IF images of sciatic nerve cross sections from WT and $Prnp^{ZH3/ZH3}$ mice at 5 and 10 d.p.c. showed upregulation of c-Jun. Nuclear c-Jun expression was faint in the uninjured sciatic nerve of a WT mouse. Channels for c-Jun and DAPI are shown separately. Scale bar 50 μM. (b) Quantification of c-Jun positive nuclei revealed no significant differences between WT and $Prnp^{ZH3/ZH3}$ mice at 5, 16 and 30 d.p.c, nor between uninjured nerves from WT and $Prnp^{ZH3/ZH3}$ mice (two-way ANOVA, Sidak's multiple comparisons test, $p > 0.05$). (c) Western blot of lysates from crushed and uninjured (contralateral) sciatic nerves at 5, 10 and 16 d.p.c. showed a pronounced upregulation of GFAP and total c-Jun protein levels both in WT and $Prnp^{ZH3/ZH3}$ mice. Also, a slight upregulation of actin was detected in crushed nerves. (d) For quantification, GFAP and c-Jun were normalized to actin and expressed as fold change to the average level of crushed WT nerves. No statistically significant difference

was detected between WT and $Prnp^{ZH3/ZH3}$ mice at the investigated time points (two-way ANOVA, Sidak's multiple comparisons test, $p > 0.05$). (e) Western blot detecting low GFAP levels in uninjured sciatic nerves from WT ($n = 6$) and $Prnp^{ZH3/ZH3}$ mice ($n = 6$) sacrificed at 5, 16 and 30 d.p.c. GFAP levels were higher in $Prnp^{ZH3/ZH3}$ sciatic nerves when compared to WT nerves. Blotting with POM2, a monoclonal anti-PrP antibody, detected PrP in WT but not in $Prnp^{ZH3/ZH3}$ sciatic nerves, confirming the absence of PrP in $Prnp^{ZH3/ZH3}$ mice.

In the early stages after nerve crush, repair Schwann cells are the main effectors of myelin breakdown and digestion [32]. Using electron microscopy (EM) we investigated the extent of demyelination by counting the number of myelin profiles that still appeared intact at 5 d.p.c. (Fig 2d and 2e). The number of intact appearing myelin profiles per sciatic nerve cross section was not significantly altered in $Prnp^{ZH3/ZH3}$ mice when compared to WT mice, indicating that the initial myelin breakdown is neither slowed nor accelerated in the absence of PrP.

Collectively, these results suggested that PrP is not required for generation and function of repair Schwann cells after nerve crush injury.

## PrP is not required for macrophage recruitment after peripheral nerve injury

Blood derived macrophages are recruited to the injured nerve via chemokines secreted by Schwann cells and mesenchymal cells [9]. Together with resident endoneurial macrophages, they contribute to debris clearance and repair after injury [33]. We performed IF for the macrophage marker CD68 to investigate if PrP is playing a role in macrophage recruitment (Fig 3a and 3b). While uninjured nerves were basically devoid of CD68-positive cells (Fig 3b), crushed sciatic nerves were infiltrated by numerous macrophages (Fig 3a). In both genotypes, macrophages were rounded and displayed a phagocytic, foamy appearance. We quantified the area covered by CD68-positive cells in relation to the S100-positive area (Fig 3c). Macrophage accumulation in the endoneurium displayed a time course which is consistent with published findings [33, 34] and showed no significant difference between WT and $Prnp^{ZH3/ZH3}$ sciatic nerves at any of the investigated time points.

Staining for Iba1, another macrophage marker, revealed a picture similar to CD68-staining (Fig 3d) and there was no significant difference in the number of Iba1-positive macrophages between WT and $Prnp^{ZH3/ZH3}$ mice at 10 d.p.c. (Fig 3e).

In addition to macrophages, the endoneurial inflammatory infiltrate comprised some CD45-positive lymphocytes with no statistically significant difference between WT and $Prnp^{ZH3/ZH3}$ mice at 10 d.p.c. (Fig 3d and 3f). These results suggest that the lack of PrP does not affect injury-induced recruitment of macrophages and lymphocytes to the peripheral nerves.

## Remyelination is not impaired in $Prnp^{ZH3/ZH3}$ mice

To investigate if PrP is required in later stages of nerve repair, we assessed remyelination after nerve crush injury by EM. We first analysed the crushed sciatic nerves at 12 d.p.c., when regrown axons are undergoing remyelination by redifferentiated Schwann cells (Fig 4). EM analysis revealed robust remyelination in both WT and $Prnp^{ZH3/ZH3}$ nerves with no statistically significant difference between genotypes (WT 85.88 ± 1.39 (3), $Prnp^{ZH3/ZH3}$ 88.56 ± 1.28 (3), percent of remyelinated axons; mean ± SEM (n), $p = 0.2282$; unpaired t-test). No gross morphological differences were noted between genotypes. At 30 d.p.c. some myelin debris was still present both in WT and $Prnp^{ZH3/ZH3}$ mice (Fig 5a and 5d). The amount of remaining debris was similar in both genotypes (WT 2.62 ± 0.37 (3), $Prnp^{ZH3/ZH3}$ 2.55 ± 0.29 (3), percentage of

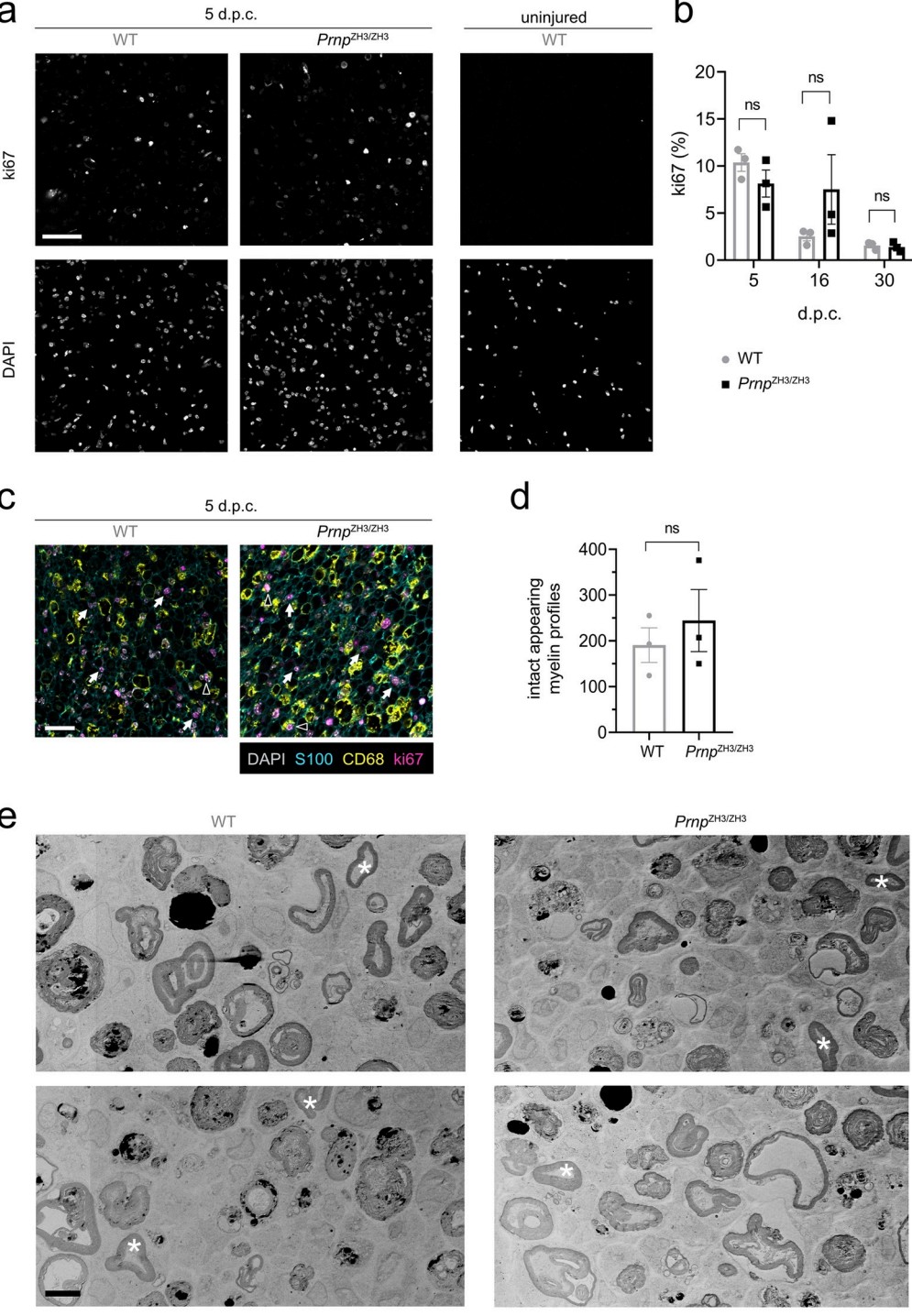

**Fig 2. Proliferation and demyelination in the early stages after crush injury.** (a) IF images of sciatic nerve cross sections from WT and $Prnp^{ZH3/ZH3}$ mice at 5 d.p.c. revealed many ki67-positive proliferating cells. The contralateral uninjured nerve from a WT mouse showed no proliferating cells. Channels for ki67 and DAPI are shown separately. Scale bar: 50 μM. (b) Ki67-proliferation index (% ki67-positive nuclei) as quantified from IF images was highest at 5 d. p.c. and decreased at later stages post crush injury. No significant differences were detected between WT and $Prnp^{ZH3/ZH3}$ mice at 5, 16 and 30 d.p.c. (two-way ANOVA, Sidak's multiple comparisons test, $p > 0.05$). n.s. = not significant. (c) IF images from crushed sciatic nerves at 5 d.p.c. stained for ki67, S100, CD68 and DAPI. The majority of ki67-positive nuclei colocalize with S100-positive Schwann cells (white arrows), whereas only few CD68-positive macrophages have ki67-positive nuclei (white triangles). Scale bar 25 μM. (d) Quantification of intact appearing myelin

profiles at 5 d.p.c. from EM images revealed no significant difference between WT and $Prnp^{ZH3/ZH3}$ mice (WT 191 ± 38 (*3*); $Prnp^{ZH3/ZH3}$ 244 ± 68 (*3*); mean ± SEM (*n*); $p$ = 0.5277; unpaired t-test). (e) EM images of sciatic nerves from WT and $Prnp^{ZH3/ZH3}$ mice at 5 d.p.c. showed extensive demyelination and few intact appearing myelin rings (white asterix). Scale bar: 5 μm.

total sciatic nerve area covered by myelin debris; mean ± SEM (*n*), $p$ = 0.8911; unpaired t-test). To determine the thickness of the myelin sheath, we performed g-ratio analysis at 10 and 30 d. p.c. In both genotypes, the g-ratio was higher in sciatic nerves at 10 d.p.c (WT 0.88 ± 0.002 (*335*), $Prnp^{ZH3/ZH3}$ 0.88 ± 0.002 (*323*), g-ratio; mean ± SEM (*number of myelinated axons quantified*), $p$ = 0.2745; unpaired t-test) than at 30 d.p.c (WT 0.73 ± 0.003 (*319*), $Prnp^{ZH3/ZH3}$ 0.72 ± 0.003 (*423*), g-ratio; mean ± SEM (*n*), $p$ = 0.4419; unpaired t-test), indicating an increase in myelin thickness. There was no significant difference in myelin thickness as assessed by g-ratio analysis between WT and $Prnp^{ZH3/ZH3}$ mice at either of the two time points (quantification for 30 d.p.c. shown in Fig 5b and 5c). Finally, we counted the total number of myelination-competent axons (with a diameter larger than 1 μm) in sciatic nerve cross sections (Fig 5e). The total number of axons was not significantly different between WT and $Prnp^{ZH3/ZH3}$ mice at 12 d.p.c. or 30 d.p.c. (12 d.p.c.: WT 3115 ± 85 (*3*), $Prnp^{ZH3/ZH3}$ 3134 ± 180 (*3*), number of axons; mean ± SEM, $p$ > 0.999. 30 d.p.c.: WT 4198 ± 33 (*3*), $Prnp^{ZH3/ZH3}$ 4245 ± 62 (*3*), number of axons; mean ± SEM, $p$ = 0.9385; one-way ANOVA, Sidak's multiple comparisons test), whereas a significant increase in axon numbers was detected in both genotypes when comparing 12 to 30 d.p.c. (p = 0.002 and p = 0.001 for WT and $Prnp^{ZH3/ZH3}$ mice, respectively).

## Discussion

Adgrg6 is expressed in Schwann cells and was shown to have autonomous and non-autonomous functions in peripheral nerve repair [12, 13]. PrP is an agonist of Adgrg6 on Schwann cells [14]. We therefore hypothesized that the lack of PrP causes similar defects in peripheral nerve repair as the knockout of Adgrg6. To evaluate our hypothesis, we performed sciatic nerve crush injuries in female WT and $Prnp^{ZH3/ZH3}$ mice and investigated the stages of repair by EM, IF and western blotting. Contrary to our expectations, we detected no differences in macrophage recruitment and remyelination when comparing sciatic nerve regeneration in $Prnp^{ZH3/ZH3}$ mice to WT mice, indicating that PrP is not responsible to induce these Adgrg6-mediated processes. Furthermore, the extent and timing of c-Jun upregulation, demyelination and Schwann cell proliferation were similar in $Prnp^{ZH3/ZH3}$ mice when compared to WT mice at all investigated time points. Our results thus suggest that PrP is not required for injury-induced peripheral nerve de- and remyelination. This may be either because PrP plays no major role in the peripheral nerve repair processes, or because other molecules can compensate for the lack of PrP. Alternatively, Adgrg6 may exhibit a basal constitutive activity sufficient to guarantee nerve repair in the absence of any agonist. Regarding the first point, PrP is dispensable for normal development of the PNS and only required for long-term maintenance of the myelin sheath [15]. Many of the repair processes after nerve injury recapitulate developmental processes [5] and it is possible that PrP is simply not required for peripheral nerve repair. As to the second point, PrP is thought to function in myelin maintenance via activation of Adgrg6 on Schwann cells. Adgrg6 is involved in peripheral nerve repair, as was suggested by deficits in macrophage recruitment and remyelination after peripheral nerve injury in conditional Adgrg6 knockout mice. Thus, Adgrg6 must sustain its function in nerve repair independent of PrP, for example via activation by other ligands such as collagen IV [35] and laminin-211 [36].

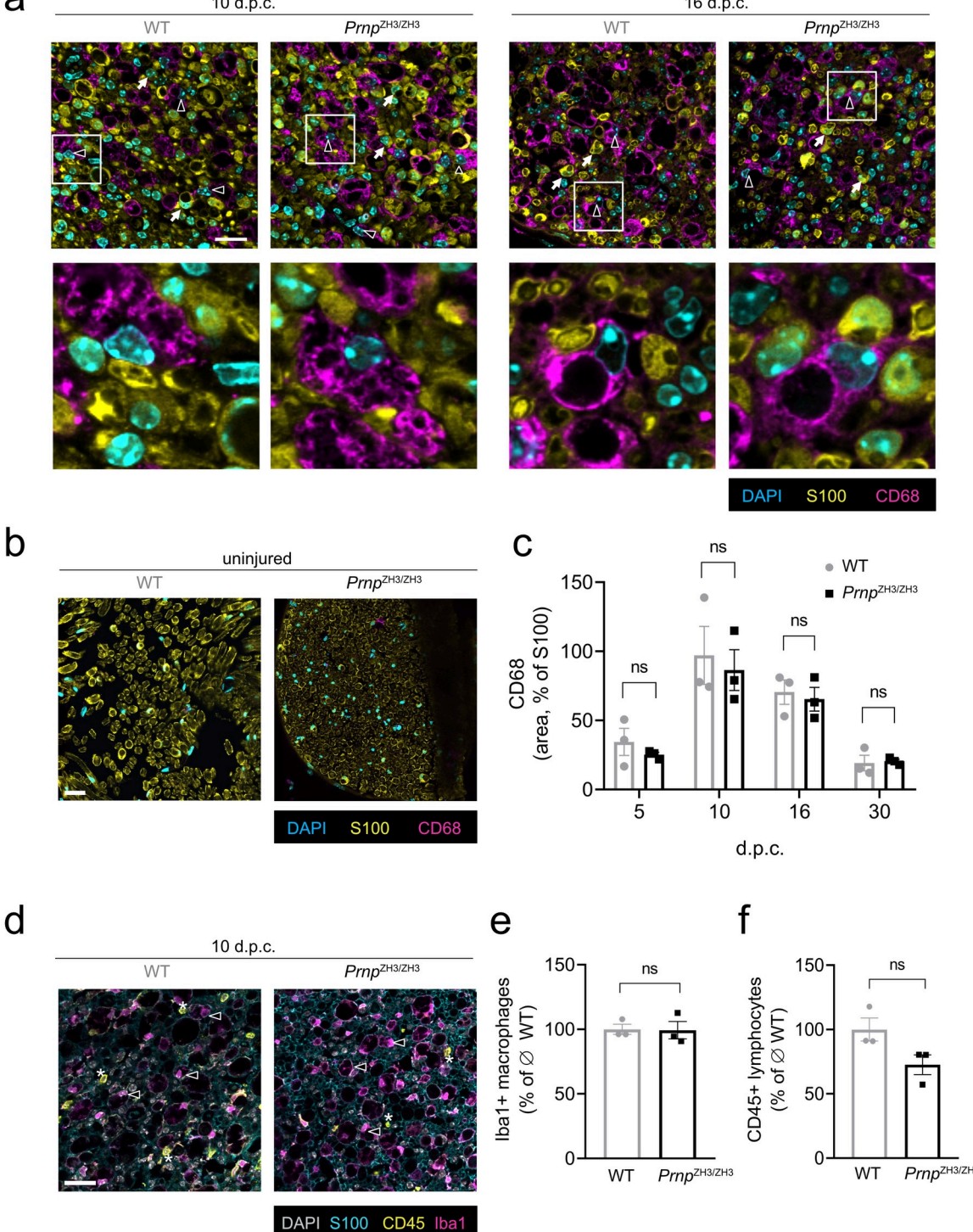

**Fig 3. Recruitment of inflammatory cells.** (a) IF images of sciatic nerve cross sections from WT and *Prnp*^ZH3/ZH3 mice at 10 and 16 d.p. c. showed pronounced infiltration by CD68-positive macrophages. White triangles: CD68-positive macrophages. White arrows: S100-positive Schwann cells. Squares indicate region selected for blown-up images of macrophages shown in the lower panels. Scale bar 25 μM. (b) IF images of uninjured (contralateral) sciatic nerve showed no infiltration by macrophages. Scale bar 25 μM. (c) The extent of macrophage infiltration (quantified as the ratio of the CD68-positive area to the S100-positive area in IF images) was not significantly different when comparing WT and *Prnp*^ZH3/ZH3 mice at all investigated time points (two-way ANOVA, Sidak's multiple comparisons test, $p > 0.05$). (d) IF images of sciatic nerve cross section from WT and *Prnp*^ZH3/ZH3 mice at 10 d.p.c showed infiltration by Iba1-positive

macrophages (white triangles) and CD45-positive lymphocytes (white asterix). Scale bar 25 μM. (e) The number of macrophages as quantified from IF images was not significantly different when comparing WT and $Prnp^{ZH3/ZH3}$ mice (WT 100 ± 3.90 (3); $Prnp^{ZH3/ZH3}$ 99.32 ± 6.80 (3); percentage of average macrophage number in WT; mean ± SEM (n); p = 0.9352; unpaired t-test). (f) The number of lymphocytes as quantified from IF images was not significantly different when comparing WT and $Prnp^{ZH3/ZH3}$ mice at 10 d.p.c. (WT 100 ± 8.88 (3); $Prnp^{ZH3/ZH3}$ 72.63 ± 7.69 (3); percentage of average lymphocyte number in WT; mean ± SEM (n); p = 0.0803; unpaired t-test). n.s. = not significant.

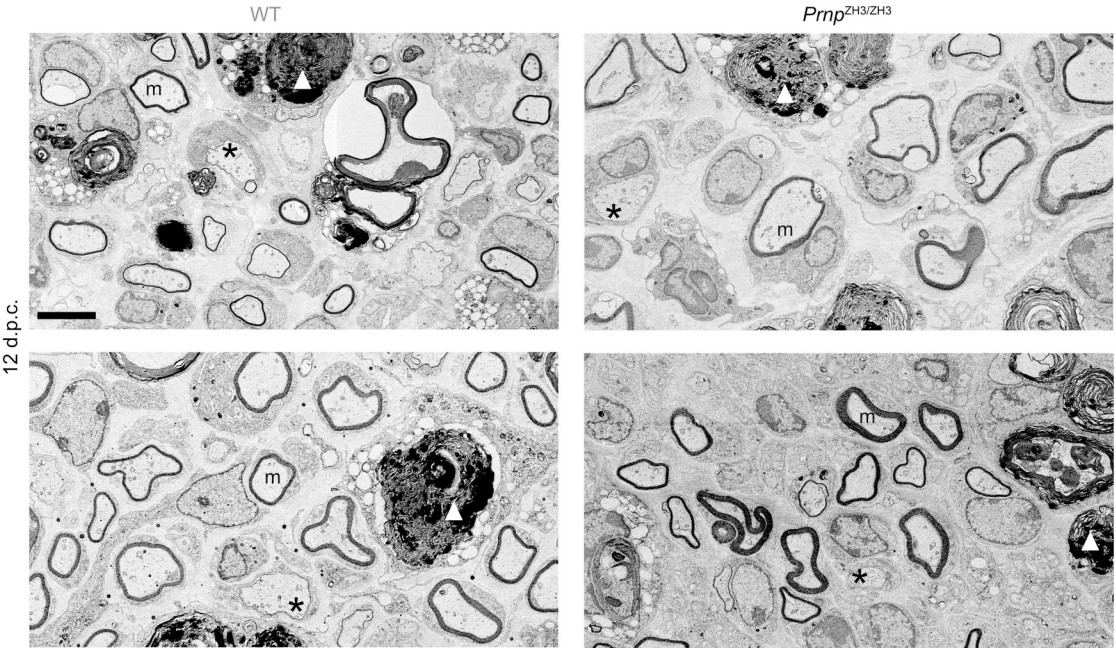

**Fig 4. Remyelination at 12 d.p.c.** EM images of sciatic nerves from WT and $Prnp^{ZH3/ZH3}$ mice at 12 d.p.c. revealed robust remyelination in both genotypes. Scale bar 5 μm.

In a further possible scenario, PrP might play contrasting roles in different cells types during the repair processes. Theoretically, the effects of PrP in different cell types may cancel each other out in a transgenic mouse with global PrP knockout as was used in the present study. However, considering that the peripheral nerve repair processes are regulated by a highly orchestrated, balanced interaction between many cell types, we feel that this is an unlikely scenario.

We have previously described that $Prnp^{ZH3/ZH3}$ mice exhibit an early upregulation of repair Schwann cell markers in sciatic nerves, which probably represents an early sign of the peripheral nerve disease [31]. In the present study, we confirmed that the 2–3 months old $Prnp^{ZH3/ZH3}$ mice undergoing crush injury showed a two-fold upregulation of GFAP in their uninjured sciatic nerves when compared to WT mice. The absolute GFAP levels after crush injury were similar in both genotypes, and the elevated baseline GFAP level in $Prnp^{ZH3/ZH3}$ mice did not interfere with regeneration. Whereas defective Schwann cell proliferation and delayed regeneration after nerve crush injury have been described in GFAP knockout mice [30, 37], the effect of GFAP overexpression on PNS development, maintenance or repair has not been investigated [38, 39]. GFAP gain-of-function mutations cause Alexander disease, a rare astrogliopathy characterized by accumulation of GFAP in the central nervous system [40]. Development of a peripheral demyelinating neuropathy was reported in an 8-year-old patient with Alexander

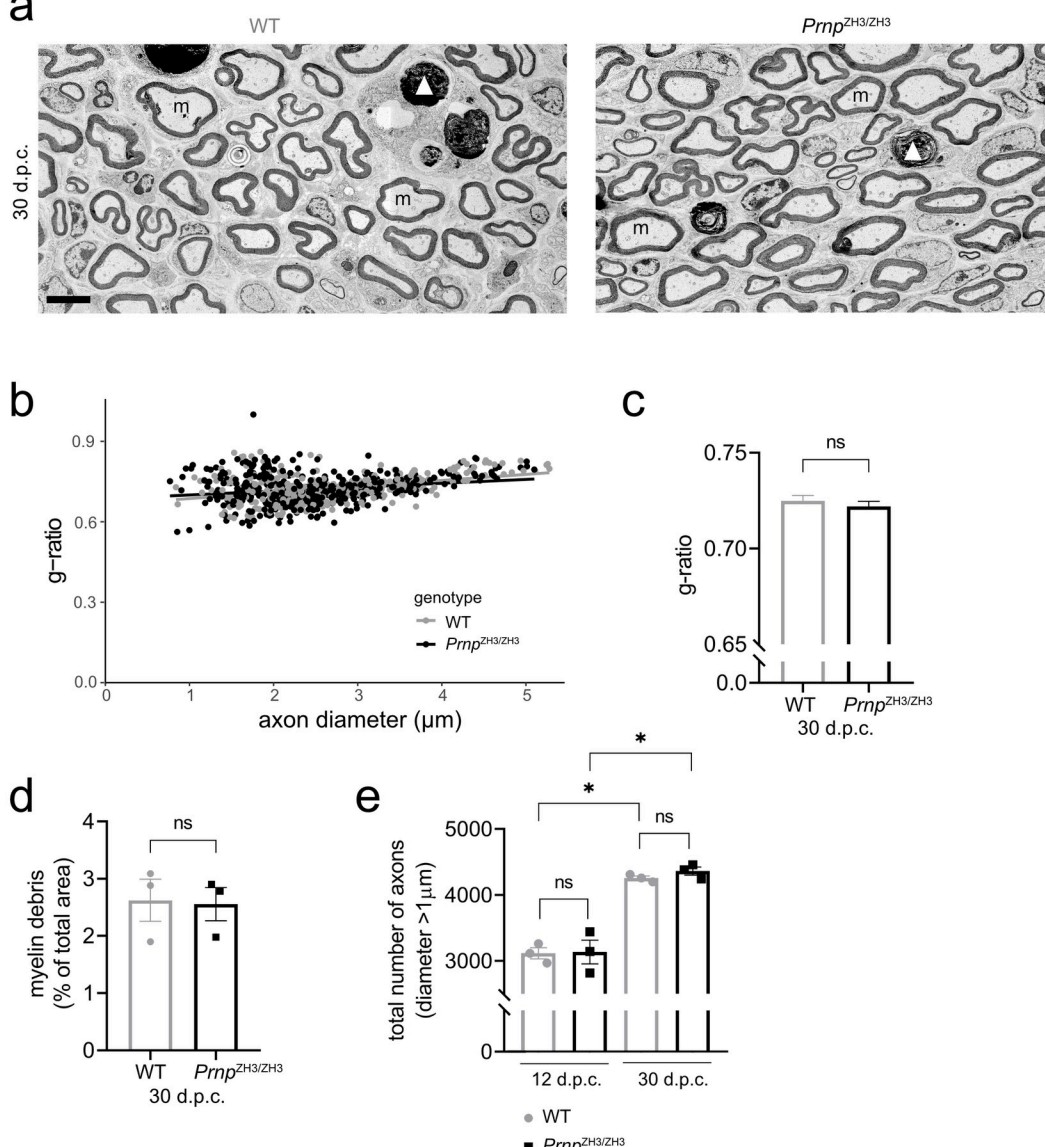

**Fig 5. Remyelination at 30 d.p.c.** (a) EM images of sciatic nerves from WT and $Prnp^{ZH3/ZH3}$ mice at 30 d.p.c. showed remyelinated axons and little myelin debris remaining. m: myelinated axons, *: non-myelinated axon with > 1 µm diameter, white arrowhead: myelin debris. Scale bar 5 µm. (b, c) G-ratio analysis in WT ($n = 3$) and $Prnp^{ZH3/ZH3}$ ($n = 3$) mice at 30 d.p.c. Quantification of > 100 myelinated axons per mouse. Straight lines in (b) show linear regression. No significant difference in mean g-ratio (c) was detected between genotypes (unpaired t-test, $p > 0.05$). (d) Quantification of area covered by myelin debris at 30 d.p.c. showed no significant difference between WT ($n = 3$) and $Prnp^{ZH3/ZH3}$ ($n = 3$) mice (unpaired t-test, $p > 0.05$). (e) The total number of myelination competent axons with diameter > 1µm increased from 12 d.p.c. to 30 d.p.c. No significant difference was detected between WT and $Prnp^{ZH3/ZH3}$ mice at 12 or 30 d.p.c. ($p > 0.05$; one-way ANOVA, Sidak's multiple comparisons test) *: $p < 0.05$. n.s. = not significant.

disease [41], but apart from this case, there is currently no compelling evidence that GFAP accumulation or overexpression contributes to or causes PNS degeneration. The finding that sciatic nerve repair is not impaired in $Prnp^{ZH3/ZH3}$ mice despite increased baseline GFAP levels might be interpreted as an indication that a baseline elevation in GFAP is still compatible with peripheral nerve regeneration.

## Limitations

While we provide molecular and morphological analyses of injury-induced Schwann cell transdifferentiation and proliferation, de- and remyelination as well as recruitment of inflammatory cells, we have not investigated other aspects of peripheral nerve repair such as axon regeneration and restoration of neuromuscular junctions. We quantified the total number of axons in sciatic nerves distal to the crush site and detected a robust increase in the number of large axons (diameter > 1 μm), indicating successful regeneration. However, we did not analyse the regeneration of axons at more distal sites nor the reinnervation of neuromuscular junctions after injury. Previous studies showed impaired reinnervation of neuromuscular junctions and defects in terminal Schwann cell responses in Adgrg6 deficient mice [12, 13]. We did not investigate whether PrP is involved in these processes. Furthermore, no functional analyses such as toe spreading index, walking track analysis or electrophysiological investigations were performed in the present study. The lack of morphological and molecular deficits at various time points post crush does not rule out (transient) differences in functional recovery between WT and $Prnp^{ZH3/ZH3}$ mice.

Our study was performed with a small number of animals ($n$ = 3 per genotype and time point). By performing the analyses at multiple time points and by combining multiple complementary readouts we aimed to increase the validity of our results. While the small $n$ warrants for caution when interpreting the results of the study, the fact that all analyses consistently showed no significant effect of PrP strengthens our conclusion. Furthermore, we are confident that a regeneration deficit of the magnitude detected in Adgrg6 knockout mice [12] should not have remained undetected in our study. Smaller defects, as well as transient functional deficits or impairment of neuromuscular junction restoration may however still be present in $Prnp^{ZH3/ZH3}$ mice.

## Conclusions

While the results of our study do not support the hypothesis that PrP is required to activate Adgrg6-mediated macrophage recruitment and remyelination after nerve crush injury, they do not rule out an ancillary, non-essential role for such interactions. Furthermore, it is possible that additional, hitherto undiscovered Adgrg6 ligands exist. The importance of such ligands to peripheral-nerve repair may vary in different species. Therefore, the negative results reported here should not discourage researchers from exploring Adgrg6 activation as a possible adjuvant therapy for peripheral nerve repair.

## Supporting information

**S1 Fig. Nerve harvest after crush injury.** The sciatic nerve was crushed using a forceps at the sciatic notch. For harvesting, the nerve was cut at 3 mm distal to the crush side. The proximal segment was embedded for electron microscopy (EM), the distal segment was used for immunofluorescence (IF) and western blotting.
(TIFF)

**S2 Fig. Original western blot images.** The uncropped blots have been inverted using the Affinity Photo software. Molecular weight markers used are indicated in kilo Dalton (kD). The arrows indicate the order in which the blots were incubated with primary antibodies. Stripping was performed before each incubation. The dashed line in blot Fig 1e indicates that the membrane was cut, and the upper and lower half were incubated with different antibodies.
(TIF)

**S3 Fig. Original immunofluorescence images.** The separate channels for each image are shown without alterations of contrast or brightness and no cropping.
(TIF)

**S4 Fig. Original immunofluorescence images.** The separate channels for each image are shown without alterations of contrast or brightness and no cropping.
(TIF)

**S1 Table. Raw data used for graphs and calculations.**
(XLSX)

## Acknowledgments

We thank Dr. Jorge A. Pereira for teaching, supervision and technical help with EM imaging, Marigona Imeri and Irina Abakumova for their valuable technical help with revision experiments, and Alexander Henzi for support with R programming and statistical analyses.

## Author Contributions

**Conceptualization:** Anna Henzi, Adriano Aguzzi.

**Data curation:** Anna Henzi.

**Formal analysis:** Anna Henzi.

**Funding acquisition:** Adriano Aguzzi.

**Investigation:** Anna Henzi, Adriano Aguzzi.

**Methodology:** Anna Henzi.

**Project administration:** Adriano Aguzzi.

**Resources:** Adriano Aguzzi.

**Supervision:** Adriano Aguzzi.

**Validation:** Anna Henzi, Adriano Aguzzi.

**Visualization:** Anna Henzi.

**Writing – original draft:** Anna Henzi.

**Writing – review & editing:** Anna Henzi, Adriano Aguzzi.

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
