## [Decision Letter · Decision Letter 0]

8 Dec 2020

PONE-D-20-33989

The prion protein is not required for peripheral nerve repair after crush injury

PLOS ONE

Dear Dr. Aguzzi,

Thank you for submitting your manuscript to PLOS ONE. After careful consideration, we feel that it has merit but does not fully meet PLOS ONE’s publication criteria as it currently stands. Therefore, we invite you to submit a revised version of the manuscript that addresses the points raised during the review process.

We look forward to receiving your revised manuscript.

Kind regards,

Simone Di Giovanni

Academic Editor

PLOS ONE

Journal Requirements:

2.PLOS ONE now requires that authors provide the original uncropped and unadjusted images underlying all blot or gel results reported in a submission’s figures or Supporting Information files. This policy and the journal’s other requirements for blot/gel reporting and figure preparation are described in detail at https://journals.plos.org/plosone/s/figures#loc-blot-and-gel-reporting-requirements and https://journals.plos.org/plosone/s/figures#loc-preparing-figures-from-image-files. When you submit your revised manuscript, please ensure that your figures adhere fully to these guidelines and provide the original underlying images for all blot or gel data reported in your submission. See the following link for instructions on providing the original image data: https://journals.plos.org/plosone/s/figures#loc-original-images-for-blots-and-gels.

Reviewers' comments:

Reviewer's Responses to Questions

**Comments to the Author**

1. Is the manuscript technically sound, and do the data support the conclusions?

Reviewer #1: Partly

Reviewer #2: Partly

2. Has the statistical analysis been performed appropriately and rigorously? 

Reviewer #1: No

Reviewer #2: Yes

3. Have the authors made all data underlying the findings in their manuscript fully available?

Reviewer #1: Yes

Reviewer #2: Yes

4. Is the manuscript presented in an intelligible fashion and written in standard English?

Reviewer #1: Yes

Reviewer #2: Yes

5. Review Comments to the Author

Reviewer #1: Schwann cell-specific loss of GPR126/Adgrg6 (for which PrPc is one interactor) results in a variety of defects in repair after peripheral nerve injury. In this work, Henzi and Aguzzi test the hypothesis that PRPc interactions with GPR126/Adgrg6 are responsible for some of these effects on peripheral nerve repair, by investigating the time course of demyelination and remyelination of the mouse sciatic nerve after crush injury in wt and PrPc null mice (PrnpZH3/ZH3). Examining demyelination, generation of “repair” Schwann cells, recruitment of macrophages to the injured nerve, and remyelination of axons 3mm from the crush site, they found no differences between wt and PrPc null animals up to 30 days after injury (except for a baseline difference in Schwann cell GFAP expression). The hypothesis tested is reasonable and clearly stated, and the data are carefully acquired and (mostly) clearly presented. The authors’ conclusion, that PrPc is not required for those aspects of repair that they studied, seems likely to be valid. For me, however, there are two main issues that should be addressed in order for this manuscript to be accepted. First, some experiments suffer from a lack of statistical power and other design/presentation issues, and second, a major aspect of nerve repair (axonal regeneration and NMJ re-formation) goes completely unexamined.

Issues with study design:

1) Why did the authors choose females and not males? Why use two months old animals rather than adult (3-6 months) mice? (3 months old mice were used for the GPR126 study)

2) There are in general 3 animals/group; was this value of N chosen based on power analyses? If so, what were the effect sizes, variances, and desired power used in the analyses? This low value of N has implications for their conclusions. Some examples follow.

3) It is clear that they failed to find differences in Figure 1, but the variance is sufficiently high that one would guess a substantially higher N would need to be sampled to detect differences smaller than a few-fold; this is also true for Figure 2b (KI67 data).

4) In Figure 3, the idea that macrophage recruitment peaked at 10 days postcrush cannot be supported by the data presented; it appears that days 10 and 16 are indistinguishable and may (or may not) be different from day 5 and day 30, but these statistical comparisons are not made.

5) If the authors decide not to perform further experiments, they should perform power analyses and provide the results in the paper, so that the strength of their conclusions can be judged.

6) Minor note: I suggest plotting 95% confidence intervals rather than SEMs, which are not really meaningful for N=3.

7) Figure S2. The main purpose of providing whole blots for examination is to judge the degree to which one can be confident of antibody performance. This is quite difficult with the current figure. As one example, the calnexin blot shows a marked calnexin band near the bottom of the gel, even though this is a 90 kD protein. The entire gels should be shown, and the positions of all MW markers demarcated with arrows, rather than using an asterisk to denote the general area of the blot where the band of interest is said to migrate. Additionally, for reviewer purposes, these blots should be shown after incubation with a single primary antibody, not multiple antibodies. Finally, the methods used to validate the antibodies used are not described. A statement is made that the asterisks mark “specific” bands, but no justification is provided for the apparent assumption that bands migrating at different molecular weights from the asterisks are “non-specific” as opposed to specific but non-desired.

Missing study elements:

8) The loss of GPR126/Adgrg6 function, specifically in Schwann cells, results in non-cell autonomous defects in axon regeneration and neuromuscular synapse formation. Clearly, axon regeneration is a sine qua non for peripheral nerve repair. However, neither axon regeneration nor NMJ reformation were examined in the current manuscript. Therefore, the statement that “PrP is dispensable for peripheral nerve repair” cannot be supported by the data provided. These experiments should be performed (ideally) or the conclusion statements should be modified.

Reviewer #2: Major points

1.) previous results by the same group (e.g. Nuvolone et al 2016) are not sufficiently mentioned and discussed in the light of the new data

2.) the mouse model is not adequately introduced: I assume it is a constitutive mouse mutant with PrP deletion in all cell types? Where are deletion phenotypes expected? Neurons and/or Schwann cells? This has to be discussed carefully.

3.) Data frequently rely on only single markers, e.g. c-Jun used for repair Schwann cells. Other markers should be used to confirm this result. Same is true for inflammation markers: here e.g. Iba1 and CD45 could also be used

4.) The authors state that GFAP is used as a Schwann cell marker, a statement not referenced and not entirely true. Typically, it is used as marker for astrocytes.

5.) The authors show altered GFAP expression between wt and mutant for the baseline. An alternative explanation to the one given by the authors might be that Prp suppresses neuroinflammation in wt mice. Please comment.

6.) For the Ki67 no co-staining with cellular markers is provided, so it is unclear in which cell type cell proliferation occurs. Co-localization experiments should be performed.

7.) For the CD68 marker ,authors previously (Nuvolone et al) described differences between wt and ko in baseline (Fig. 3b). Why is this not observed in the current study?

8.) The authors do not employ any functional assays to monitor nerve regeneration which limits their data interpretation. This limitation should be clearly stressed.

9.) Results and quantification in Fig. 1e seem mainly to rely on unequal loading as depicted by calnexin. Please comment.

10.) Numbers of animals at each timepoint are rather low (e.g. N=3). For instance in Fig. 2b there are 3 animals with one clear outlier. However, I see the point that authors include several timepoints. Nevertheless, low N-number should be discussed.

11.) No quantification for data in Fig. 2c is provided which should be included.

Minor points

1.) the introduction is overall very short and does not provide sufficient background

2.) In the figures author should designate mutant mice with mutant rather than simply ZH3

3.) Fig. 1b should also include uninjured even if values are zero

6. PLOS authors have the option to publish the peer review history of their article (what does this mean?). If published, this will include your full peer review and any attached files.

Reviewer #1: No

Reviewer #2: No

---

## [Author Response · Author response to Decision Letter 0]

21 Dec 2020

Response to Reviewers

Rebuttal letter [PONE-D-20-33989]

We thank the reviewers for their very careful and thorough assessment of our study. By performing additional analyses as well as altering the statements in our manuscript and discussing the limitations of our study in more depth we hope to meet the requirements of the reviewers.

The original western blots and images shown in the manuscript are provided in Supporting information, which is also stated in the Methods section. We now additionally provide a table containing all the raw data underlying the graphs and statistical analyses.

We used the PACE tool for converting our images to the required format.

Please find our responses to the individual remarks below. The lines indicating the changes refer to the manuscript with track changes.

Reviewer #1: 

1) Why did the authors choose females and not males? Why use two months old animals rather than adult (3-6 months) mice? (3 months old mice were used for the GPR126 study)

Female mice were chosen for practical reasons: After the surgery, male mice are more prone to wound reopening than female mice, and they cannot be regrouped after a period of single housing to allow for wound healing. To avoid loss of animals in the experiment due to repeated wound reopening (which would require euthanasia according to our SOP) and to avoid prolonged single housing of animals, we performed the experiment with female mice only. These explanations have been integrated in the Methods section of the manuscript (lines 79-82 in the manuscript with track changes). 

As to the age of the mice: the peripheral demyelinating disease developed by PrP knockout mice could interfere with remyelination, and we would not be able to distinguish this from an effect caused by the mere lack of PrP. To avoid confounding effects by impaired PrP-based myelin maintenance, we performed the experiments in mice at the youngest age possible (after completion of peripheral myelination). We included this explanation in the text (lines 199-203).

2) There are in general 3 animals/group; was this value of N chosen based on power analyses? If so, what were the effect sizes, variances, and desired power used in the analyses? This low value of N has implications for their conclusions. Some examples follow.

Sample sizes are similar to those of previous publications performing western blot, IF and EM analyses in sciatic nerve crush experiments [1–3]. Furthermore, they are in accordance with a sample size calculation based on the results from the CD68- and Iba1-immunohistochemistry studies reported in Mogha et al, J Neurosci, 2016 [1] (calculated with the G*power software version 3.1.9.7 with power 0.8, type I error rate of 5%, effect sizes ranging from 2-6 in [1]). Mogha et al. used an inducible knockout model. We expected less variability in our mouse model due to the complete knockout of PrP in contrast to the tamoxifen-induced Adgrg6 knockout in [1].

By performing the analyses at multiple time points and combining complementary methods, we aimed to further increase the validity of our conclusion. For the TEM studies, the acquisition and analysis of the whole sciatic nerve cross section provides a very precise and comprehensive readout, which increases the strength of our conclusion. In the extensive experience of our collaborator with nerve crush experiments, N = 3 is sufficient to detect clinically meaningful differences between groups – in particular when multiple time points are analysed [2,3]. 

We have adapted the respective section in Material and Methods accordingly and we included a discussion about the small sample size in lines 456-464.

3) It is clear that they failed to find differences in Figure 1, but the variance is sufficiently high that one would guess a substantially higher N would need to be sampled to detect differences smaller than a few-fold; this is also true for Figure 2b (KI67 data).

Please refer to the response above. 

4) In Figure 3, the idea that macrophage recruitment peaked at 10 days postcrush cannot be supported by the data presented; it appears that days 10 and 16 are indistinguishable and may (or may not) be different from day 5 and day 30, but these statistical comparisons are not made.

We agree with the comment of the reviewer. Our data do not necessarily show that the number of macrophages peaks at 10 days post crush. We have altered the text accordingly. We did not perform additional comparisons since the exact peak of macrophage recruitment is not relevant for the conclusion that no difference can be detected between WT and PrP knockout mice. We have adapted the text accordingly (lines 315 and 333).

5) If the authors decide not to perform further experiments, they should perform power analyses and provide the results in the paper, so that the strength of their conclusions can be judged.

We have now provided the results of a power analysis based on Mogha et al, J Neurosci, 2016 [1] in the material and methods section. A post hoc power analysis of our own results does not provide any further insights [4]. We have added a paragraph to discuss the small sample size in the discussion (lines 456-464).

6) Minor note: I suggest plotting 95% confidence intervals rather than SEMs, which are not really meaningful for N=3.

We compared the graphs using CI or SEM for the error bars (see example in the attached "Response to Reviewers" Word document). When displayed in the figures, the shorter error bars representing the SEM (compared to the longer error bars for CI) enable a better discrimination of the individual points, which we are plotting for all experiments (except for the summary of the g-ratio). Therefore, we prefer to plot SEM in the figures. We argue that no relevant information is lost by plotting SEM instead of CI, 1) since we are plotting the individual data points and 2) the CI is directly related to SEM by t statistics. Furthermore, we are now providing all the raw data underlying the graphs in Supplementary Table S1. This will allow readers to reproduce and/or extend the statistical analysis.

7) Figure S2. The main purpose of providing whole blots for examination is to judge the degree to which one can be confident of antibody performance. This is quite difficult with the current figure. As one example, the calnexin blot shows a marked calnexin band near the bottom of the gel, even though this is a 90 kD protein. The entire gels should be shown, and the positions of all MW markers demarcated with arrows, rather than using an asterisk to denote the general area of the blot where the band of interest is said to migrate. Additionally, for reviewer purposes, these blots should be shown after incubation with a single primary antibody, not multiple antibodies. Finally, the methods used to validate the antibodies used are not described. A statement is made that the asterisks mark “specific” bands, but no justification is provided for the apparent assumption that bands migrating at different molecular weights from the asterisks are “non-specific” as opposed to specific but non-desired.

We have repeated the western blots without cutting the membranes. All molecular weight markers are now indicated on the images. We have performed stripping before subsequent incubations and described the procedure in the material and methods section. The unprocessed raw blots are provided in Supplementary Figure S2.

The antibodies were considered valid if a strong, consistent band was detected at the expected molecular weight. Additionally, the antibodies used in our study are commonly used in our laboratory for both western blotting and immunostaining. We have also used them in a previous study about the peripheral demyelination in Prnp knockout mice [5]. 

Missing study elements

8) The loss of GPR126/Adgrg6 function, specifically in Schwann cells, results in non-cell autonomous defects in axon regeneration and neuromuscular synapse formation. Clearly, axon regeneration is a sine qua non for peripheral nerve repair. However, neither axon regeneration nor NMJ reformation were examined in the current manuscript. Therefore, the statement that “PrP is dispensable for peripheral nerve repair” cannot be supported by the data provided. These experiments should be performed (ideally) or the conclusion statements should be modified.

We did not collect muscle tissue from the mice in our study. Thus, we unfortunately cannot perform an analysis of NMJ reformation without repeating the entire experiment, which we decided to be out of scope for the present study. We agree that NMJ reinnervation is an important part of peripheral nerve regeneration. We have altered our statements in the introduction and the discussion accordingly. 

For axon regeneration, we now included a quantification of the total number of myelination-competent axons (i.e. diameter larger than 1um, in TEM) at 12 d.p.c and 30 d.p.c. as a rough measure of axon regeneration. The results are now included in lines 365-373 and Fig 5e. We also discuss axon regeneration in the limitations section of the discussion.

Reviewer #2: Major points

1.) previous results by the same group (e.g. Nuvolone et al 2016) are not sufficiently mentioned and discussed in the light of the new data

We have extended both the introduction and the discussion section to better represent the results of previous studies.

2.) the mouse model is not adequately introduced: I assume it is a constitutive mouse mutant with PrP deletion in all cell types? Where are deletion phenotypes expected? Neurons and/or Schwann cells? This has to be discussed carefully.

We have now better introduced the mouse model in the manuscript (lines 64 and 199-203).

3.) Data frequently rely on only single markers, e.g. c-Jun used for repair Schwann cells. Other markers should be used to confirm this result. Same is true for inflammation markers: here e.g. Iba1 and CD45 could also be used

For detection of repair Schwann cells, we used c-Jun and GFAP. Furthermore, quantification of early demyelination in TEM as well as Schwann cell proliferation can also be considered as indicators of functional repair Schwann cells. Thus, we consider our characterization of repair Schwann cells to be sufficient.

For recruitment of inflammatory cells, we have now performed additional immunofluorescent stainings at one of the time points (10 d.p.c.). The results are included in the manuscript (lines 318-320 and Figure 3d). 

4.) The authors state that GFAP is used as a Schwann cell marker, a statement not referenced and not entirely true. Typically, it is used as marker for astrocytes.

While GFAP labels astrocytes (and, surprisingly, some cerebellar neuronal populations and some meningeal cells) in the central nervous system, it is also a reliable marker of repair Schwann cells in the peripheral nervous system. We now provide more background information to justify our choice of GFAP as a marker of repair Schwann cells (lines 249-252).

5.) The authors show altered GFAP expression between wt and mutant for the baseline. An alternative explanation to the one given by the authors might be that Prp suppresses neuroinflammation in wt mice. Please comment.

GFAP expression in PrP knockout mice is detectable at an age before infiltration by macrophages occurs and other morphological signs of demyelination can be detected [5,6]. We do interpret the increase in GFAP expression as an early sign of nerve damage occurring due to the absence of PrP. We currently do not know whether the damage is related to inflammatory processes. In our previous study [7], the demyelinating disease could not be suppressed in Rag-knockout mice, suggesting that B and T cells do not contribute to the pathogenesis.

6.) For the Ki67 no co-staining with cellular markers is provided, so it is unclear in which cell type cell proliferation occurs. Co-localization experiments should be performed.

We now performed a double staining with CD68, S100 and ki67 from nerves at 5 d.p.c. The results are now included in lines 270-275 and Figure 2c.

7.) For the CD68 marker ,authors previously (Nuvolone et al) described differences between wt and ko in baseline (Fig. 3b). Why is this not observed in the current study?

In Nuvolone et al, a significant difference in the presence of CD68 positive macrophages in the sciatic nerve was only observed at 9 months of age (and a trend towards increased macrophage numbers at 3 months of age). Thus, we do not expect a significant difference in the number of macrophages at 2 months of age. This also contributed to our decision of performing the crush experiments at 2 months of age. We aimed to avoid any interference with regeneration by increased numbers of macrophages in ZH3 mice at baseline. We now mention this in lines 199-200.

8.) The authors do not employ any functional assays to monitor nerve regeneration which limits their data interpretation. This limitation should be clearly stressed.

We have added a “limitations” section to the discussion (lines 441-464), where we discuss the lack of functional assays in our study.

9.) Results and quantification in Fig. 1e seem mainly to rely on unequal loading as depicted by calnexin. Please comment.

We agree with the reviewer that the loading control is uneven. Since we loaded the same amount of protein for all samples (10ug), we think that Calnexin is upregulated in sciatic nerves at 1-2 weeks post crush. We have now repeated the western blots with actin as loading control, and we also found a slight upregulation of actin at the investigated time points. To account for this, we are now presenting the protein levels as fold change to the average levels in crushed WT nerves.

10.) Numbers of animals at each timepoint are rather low (e.g. N=3). For instance in Fig. 2b there are 3 animals with one clear outlier. However, I see the point that authors include several timepoints. Nevertheless, low N-number should be discussed.

We agree that this is a limitation of our study. We are now discussing this point in the manuscript (please refer to the new “limitations” section of the discussion).

11.) No quantification for data in Fig. 2c is provided which should be included.

We have added the respective bar graph to Figure 2 and moved the descriptive statistics from the main text to the figure legend.

Minor points

1.) the introduction is overall very short and does not provide sufficient background

We have extended the introduction to introduce more background information.

2.) In the figures author should designate mutant mice with mutant rather than simply ZH3

We have changed the legends in all figures accordingly.

3.) Fig. 1b should also include uninjured even if values are zero

We have included the values in the graph.

References

[1] Mogha A, Harty BL, Carlin D, Joseph J, Sanchez NE, Suter U, et al. Gpr126/Adgrg6 Has Schwann Cell Autonomous and Nonautonomous Functions in Peripheral Nerve Injury and Repair. J Neurosci 2016;36:12351–67. https://doi.org/10.1523/JNEUROSCI.3854-15.2016.

[2] Gerber D, Ghidinelli M, Tinelli E, Somandin C, Gerber J, Pereira JA, et al. Schwann cells, but not Oligodendrocytes, Depend Strictly on Dynamin 2 Function. ELife 2019;8:e42404. https://doi.org/10.7554/eLife.42404.

[3] Norrmén C, Figlia G, Pfistner P, Pereira JA, Bachofner S, Suter U. mTORC1 Is Transiently Reactivated in Injured Nerves to Promote c-Jun Elevation and Schwann Cell Dedifferentiation. J Neurosci 2018;38:4811–28. https://doi.org/10.1523/JNEUROSCI.3619-17.2018.

[4] Zhang Y, Hedo R, Rivera A, Rull R, Richardson S, Tu XM. Post hoc power analysis: is it an informative and meaningful analysis? Gen Psych 2019;32:e100069. https://doi.org/10.1136/gpsych-2019-100069.

[5] Henzi A, Senatore A, Lakkaraju AKK, Scheckel C, Mühle J, Reimann R, et al. Soluble dimeric prion protein ligand activates Adgrg6 receptor but does not rescue early signs of demyelination in PrP-deficient mice. PLoS ONE 2020;15:e0242137. https://doi.org/10.1371/journal.pone.0242137.

[6] Nuvolone M, Hermann M, Sorce S, Russo G, Tiberi C, Schwarz P, et al. Strictly co-isogenic C57BL/6J-Prnp-/- mice: A rigorous resource for prion science. J Exp Med 2016;213:313–27. https://doi.org/10.1084/jem.20151610.

[7] Bremer J, Baumann F, Tiberi C, Wessig C, Fischer H, Schwarz P, et al. Axonal prion protein is required for peripheral myelin maintenance. Nat Neurosci 2010;13:310–8. https://doi.org/10.1038/nn.2483.

---

## [Decision Letter · Decision Letter 1]

11 Jan 2021

The prion protein is not required for peripheral nerve de- and remyelination after crush injury

PONE-D-20-33989R1

Dear Dr. Aguzzi,

We’re pleased to inform you that your manuscript has been judged scientifically suitable for publication and will be formally accepted for publication once it meets all outstanding technical requirements.

Kind regards,

Simone Di Giovanni

Academic Editor

PLOS ONE

Reviewer's Responses to Questions

**Comments to the Author**

1. If the authors have adequately addressed your comments raised in a previous round of review and you feel that this manuscript is now acceptable for publication, you may indicate that here to bypass the “Comments to the Author” section, enter your conflict of interest statement in the “Confidential to Editor” section, and submit your "Accept" recommendation.

Reviewer #2: All comments have been addressed

2. Is the manuscript technically sound, and do the data support the conclusions?

Reviewer #2: Yes

3. Has the statistical analysis been performed appropriately and rigorously? 

Reviewer #2: Yes

4. Have the authors made all data underlying the findings in their manuscript fully available?

Reviewer #2: Yes

5. Is the manuscript presented in an intelligible fashion and written in standard English?

Reviewer #2: Yes

6. Review Comments to the Author

Reviewer #2: The authors have addressed most concerns adequately and added words of caution when interpreting data so I am happy to suggest accepting the manuscript

7. PLOS authors have the option to publish the peer review history of their article (what does this mean?). If published, this will include your full peer review and any attached files.

Reviewer #2: No

---

## [Editor Report · Acceptance letter]

12 Jan 2021

PONE-D-20-33989R1 

The prion protein is not required for peripheral nerve de- and remyelination after crush injury 

Dear Dr. Aguzzi:

I'm pleased to inform you that your manuscript has been deemed suitable for publication in PLOS ONE. Congratulations! Your manuscript is now with our production department. 

Kind regards, 

on behalf of

Dr. Simone Di Giovanni 

Academic Editor

PLOS ONE